# A New Lichenized Fungus, *Psoroglaena humidosilvae*, from a Forested Wetland of Korea, with a Taxonomic Key to the Species of *Psoroglaena*

**DOI:** 10.3390/jof8040392

**Published:** 2022-04-12

**Authors:** Beeyoung Gun Lee, Jae-Seoun Hur

**Affiliations:** 1Baekdudaegan National Arboretum, Bonghwa 36029, Korea; 2Korean Lichen Research Institute, Suncheon National University, Suncheon 57922, Korea; jshur1@sunchon.ac.kr

**Keywords:** biodiversity, corticolous, phylogeny, taxonomy, Verrucariaceae

## Abstract

*Psoroglaena humidosilvae* Lee is described as a new lichen species from a wetland forest in South Korea. The new species is distinct from *P. stigonemoides* (Orange) Henssen by little projections locally present on the thallus; smaller, paler, and globose perithecia; smaller asci; and smaller ascospores generally 3-septate. Molecular analyses employing internal transcribed spacer (ITS), mitochondrial small subunit (mtSSU), and nuclear large subunit ribosomal RNA (LSU) sequences strongly support *P. humidosilvae* as a nonidentical species in the genus *Psoroglaena*. A surrogate key is provided to assist in the identification of all 22 species of *Psoroglaena*.

## 1. Introduction

Since the genus *Psoroglaena* was introduced in 1891 [1], the small genus (21 spp.) has been confused with analogous genera *Agonimia*, *Leucocarpia*, *Macentina*, and *Phylloblastia* in the large family Verrucariaceae (ca. 970 spp.) [2]. *Psoroglaena* is delimited by the key characteristics such as a filamentous, squamulose, granulose, or crustose thallus, mycobiont hyphae often with papillae, pale yellow, pale brown, brownish gray, or blackish perithecia which have no or a reduced involucrellum, and fusiform-ellipsoid to oblong-fusiform ascospores [3,4,5]. *Psoroglaena* differs from *Agonimia* by pale perithecia and exciples, generally eight-spored asci, and transversely septate ascospores as well as muriform ones [6]. *Psoroglaena* is distinguished from *Phylloblastia* by a filamentous thallus, mycobiont hyphae often papillose, and pale yellow to blackish perithecia having no or a reduced involucrellum, and substrate preference to bark as well as leaf, bryophyte, rock, and mossy soil [4,6,7]. Species of *Phylloblastia* have been detected only on leaf surfaces, and the taxonomy of a corticolous *P. gyeongsangbukensis* J.P. Halda, J.S. Park, and Hur may not be convincing [8]. Another two genera, *Leucocarpia* and *Macentina*, are regarded as congeneric to *Psoroglaena* [3,4]. The distribution of *Psoroglaena* is cosmopolitan from temperate, subtropical to tropical regions [5]. A few studies on molecular phylogeny including *Psoroglaena* have been accomplished [9,10] although other molecular works have been carried out at the family level without *Psoroglaena* [11,12,13,14,15] or the photobionts for *Psoroglaena* were phylogenetically classified [16,17]. Five species of *Psoroglaena* have been recorded in Korea since 2016, including *P. chirisanensis* Lőkös, S.Y. Kondr. and Hur, *P. coreana* S.Y. Kondr., Lőkös and Hur, *P. gangwondoensis* S.Y. Kondr., Lőkös, J.-J. Woo and Hur, *P. japonica* H. Harada, and *P. sunchonensis* S.Y. Kondr., Lőkös and Hur [18,19,20,21,22].

This study describes a new lichenized fungus. One of the field surveys for the lichen biodiversity in the forested wetlands of South Korea were accomplished during the summer of 2021, and five specimens in *Psoroglaena* were collected in a small swamp of a mountain (Figure 1). The specimens were analyzed in ecology, morphology, chemistry, and molecular phylogeny, and did not correspond to any previously known species. We describe them as a new species, *Psoroglaena humidosilvae*, and a taxonomic key is provided for all *Psoroglaena* species. The specimens are deposited in the herbarium of the Baekdudaegan National Arboretum (KBA), South Korea.

## 2. Materials and Methods

### 2.1. Morphological and Chemical Analyses

Hand sections were prepared manually with a razor blade under a stereomicroscope (Olympus optical SZ51; Olympus, Tokyo, Japan), scrutinized under a compound microscope (Nikon Eclipse E400; Nikon, Tokyo, Japan), and pictured using a software program (NIS-Elements D; Nikon, Tokyo, Japan) and a DS-Fi3 camera (Nikon, Tokyo, Japan) mounted on a Nikon Eclipse Ni-U microscope (Nikon, Tokyo, Japan). The ascospores were examined at 1000× magnification in water. The length and width of the ascospores were measured and the range of spore sizes was shown with average, standard deviation (SD), length-to-width ratio, and number of measured spores. Thin-layer chromatography (TLC) was performed using solvent systems A and C according to standard methods [23].

### 2.2. Isolation, DNA Extraction, Amplification, and Sequencing

Hand-cut sections of ten to twenty ascomata per collected specimen were prepared for DNA isolation and DNA was extracted with a NucleoSpin Plant II Kit in line with the manufacturer’s instructions (Macherey-Nagel, Düren, Germany). PCR amplification for the internal transcribed spacer region (ITS1-5.8S-ITS2 rDNA), the mitochondrial small subunit, and the nuclear large subunit ribosomal RNA genes was achieved using Bioneer’s AccuPower PCR Premix (Bioneer, Daejeon, Korea) in 20-μL tubes with 16 μL of distilled water, 2 μL of DNA extracts, and 2 μL of primers ITS5 and ITS4 [24], mrSSU1 and mrSSU3R [25] or LR0R and LR5 [26]. The PCR thermal cycling parameters used were 95 °C (15 s), followed by 35 cycles of 95 °C (45 s), 54 °C (45 s), and 72 °C (1 min), and a final extension at 72 °C (7 min) based on Ekman [27]. The annealing temperature was occasionally altered by ±1 degree in order to obtain a better result. PCR purification and DNA sequencing were accomplished by the genomic research company Macrogen (Seoul, Korea).

### 2.3. Phylogenetic Analyses

All ITS, mtSSU, and LSU sequences were aligned and edited manually using ClustalW in Bioedit V7.2.6.1 [28]. All ambiguous characters were removed and only unambiguous characters were analyzed. The final alignment comprised 2338 (ITS), 968 (mtSSU), and 2404 (LSU) columns. Phylogenetic trees with bootstrap values were obtained in RAxML GUI 2.0.6 [29] using the Maximum Likelihood method with a rapid bootstrap with 1000 bootstrap replications and GTR GAMMA (TrN + I + G4 for ITS, TPM3uf + G4 for mtSSU, and TIM2 + G4 for LSU) for the substitution matrix, as the best models produced by the model test in the software. The posterior probabilities were obtained in BEAST 2.6.4 [30] using the GTR 121131/TN93 (ITS), the GTR 123123 (mtSSU), and the GTR 121343 (LSU) models, as the appropriate models of nucleotide substitution produced by the Bayesian model averaging methods with bModelTest [31], empirical base frequencies, gamma for the site heterogeneity model, four categories for gamma, and a 10,000,000 Markov chain Monte Carlo chain length with a 10,000-echo state screening and 1000 log parameters. Then, a consensus tree was constructed in TreeAnnotator 2.6.4 [30] with the first 25% discard as a burn-in, no posterior probability limit, a maximum clade credibility tree for the target tree type, and median node heights. All trees were displayed in FigTree 1.4.2 [32] and edited in Microsoft Paint. The phylogenetic trees and DNA sequence alignments are deposited in TreeBASE under the study ID 29486. Most sequences employed for the analyses are based on Muggia et al. [9]. Overall analyses in the materials and methods were accomplished based on Lee and Hur [33].

## 3. Results

### 3.1. Phylogenetic Analyses

Three independent phylogenetic trees for the genus *Psoroglaena* and related genera in the family Verrucariaceae were produced from 91 sequences (27 for ITS, 33 for mtSSU, and 31 for LSU) from GenBank, and nine new sequences (3 for each locus) from the new species (Table 1). The new species is positioned in *Psoroglaena* in all ITS, mtSSU, and LSU trees. The ITS tree shows that the new species is clustered with *P. stigonemoides* and *P.* sp., represented by a bootstrap value of 100 and a posterior probability of 1.00 for the branch (Figure 2). Although the new species is neighboring with other *Psoroglaena* species in the branch, the species is separately located without coupling with any identical species. The mtSSU tree illustrates that the new species is grouped with *P. abscondita* (Coppins and Vězda) Hafellner and Türk, *P. biatorella* (Arnold) Lücking and Sérus., and *P.* sp., represented by a bootstrap value of 100 and a posterior probability of 1.0 for the branch (Figure 3). Although located closer to *P. abscondita* and *P. biatorella* than *P.* sp. in the branch, the new species is distinctly situated away from the close species. The LSU tree explains that the new species is arranged with other *Psoroglaena* species, i.e., *P. biatorella* and *P. stigonemoides*, represented by a bootstrap value of 57 (not shown) and a posterior probability of 1.00 (not shown) for the branch (Figure 4). Although the new species is grouped with the *Psoroglaena* species those are classified in a different clade separated from the new species. Unexpectedly, *Psoroglaena abscondita* is located in a clade of the genus *Placidium*, far from all other *Psoroglaena* species. This species is further considered in the discussion.

### 3.2. Taxonomy

#### 3.2.1. *Psoroglaena humidosilvae* B.G. Lee sp. nov.

MycoBank: MB 842927 (Figure 5).

Type: SOUTH KOREA, North Chungcheong Province, Jincheon, Baekgok-myeon, a forested wetland, 36°55′09.37″ N, 128°18′14.2″ E, 346 m alt., on bark of *Salix koreensis*, 2 June 2021, B.G. Lee and H.J. Lee 2021-000535 (holotype: KBA-L-0002007!; GenBank OM811989 for ITS, OM811986 for mtSSU, and OM811983 for LSU); same locality, on bark of *Salix koreensis*, 2 June 2021, B.G. Lee and H.J. Lee 2021-000528 (paratype: KBA-L-0002000; same locality, on bark of *Salix koreensis*, 2 June 2021, B.G. Lee and H.J. Lee 2021-000533 (paratype: KBA-L-0002005; GenBank OM811988 for ITS, OM811985 for mtSSU, and OM811982 for LSU); same locality, on bark of *Salix koreensis*, 2 June 2021, B.G. Lee and H.J. Lee 2021-000534 (paratype: KBA-L-0002006); same locality, on bark of *Salix koreensis*, 2 June 2021, B.G. Lee and H.J. Lee 2021-000536 (paratype: KBA-L-0002008; GenBank OM811990 for ITS, OM811987 for mtSSU, and OM811984 for LSU).

Description: Thallus corticolous, crustose, areolate in young stage and soon continuous as developed, smooth to granular, with locally appressed filaments as minute projections, branched or not branched, thin, pale greenish- or whitish-gray to green, margin indeterminate, 60–150 μm thick; cortex hyaline, 5–15 μm thick, cortical cells granular, 5–10 μm diam., with outermost 1–2 layers periclinally arranged, up to 5 μm thick; medulla 10–25 μm thick, below algal layer; photobiont coccoid, cells globose, subglobose to ellipsoid, 5–12 μm thick, algal layer 10–90 μm thick; thalline hyphae occasionally papillose, papillae containing algae when mature, 9–65 × 7–40 μm (n = 10). Small crystals present between algal cells or in cortex, dissolving in K. Prothallus absent.

Perithecia very numerous, solitary but close to each other, not contiguous, mostly covered by thalline collar and only ostiolar region shown, globose, ostiolar region 20–30 μm high and the immersed part by thalline collar 100–150 μm high, 0.12–0.22 mm diam. (mean = 0.16, SD = 0.02, n = 103), ostiolar region flat to slightly convex, pale yellow (in both dry and wet conditions) with a dark or transparent minute ostiole in the center, c. 100 μm diam. Involucrellum absent. Excipulum not carbonized, almost hyaline or slightly pale yellowish, of 5- to 6-layered periclinal hyphae, 40–50 μm thick at the ostiolar region, c. 20 μm thick laterally, 20–40 μm thick at the base, sometimes with small crystals, dissolving in K. Paraphyses absent. Periphyses irregularly septate, branched, 20–25 × 1–1.5 μm at the ostiolar region, c. 10 × 1–1.5 μm at the base.

Asci fissitunicate, 8-spored (rarely 6-spored), clavate to cylindrical, 30–65 × 10–20 μm (n = 10). Ascospores 3-septate (often 1-septate when mature), locules almost equally divided or with the middle locules larger, narrowly ellipsoid to ellipsoid-fusiform, generally without constriction at septum (slightly constricted at septum for some 1-septate spores), with thin and wavy septa, 12–18 × 4.5–6.5 μm (mean = 15.3 × 5.4 μm; SD = 1.29 (L), 0.39 (W); L/W ratio 2.2–3.6, ratio mean = 2.8, ratio SD = 0.3; n = 106), without epispore. Pycnidia not detected.

Chemistry: Thallus K–, KC–, C–, Pd–, UV–. Perithecia K–, I–, KI–. No lichen substance was detected by TLC.

Distribution and ecology: The species occurs on the bark of *Salix koreensis*. The species is currently known from the type collections.

Etymology: The species epithet indicates the lichen’s habitat, i.e., a humid forest.

Notes: The new species is most similar to *P. stigonemoides* in having projections on thallus and transversely septate ascospores among corticolous species. However, *P. stigonemoides* differs from the new species by heavy projections covered on thallus, larger, darker, and ovoid-obpyriform perithecia, larger asci, and larger ascospores up to 5-septate [34].

The new species can be compared with *P. abscondita* and *P. chirisanensis* in having transversely 3-septate ascospores among corticolous species. However, *P. abscondita* represents leprose thallus, darker perithecia, narrower exciple, and longer but narrower ascospores [34,35]. *Psoroglaena chirisanensis* differs from the new species by thallus without projections and larger ascospores with distinct constriction at septa [18] (Table 2).

#### 3.2.2. Key to the Species of *Psoroglaena*

The key is composed of all 22 species in the genus *Psoroglaena*, which disregarded two infraspecific taxa of *P. cubensis* Müll. Arg., i.e., *P. cubensis* var. *cubensis* Müll. Arg. and *P. cubensis* var. *teretiloba* O.E. Erikss. (Table 3). The key couplets for foliicolous species are referenced from Lücking [4].

## 4. Discussion

The phylogenetic results strongly support that the new species is unique in *Psoroglaena* in all ITS, mtSSU, and LSU trees. Although previously recorded DNA sequences for *Psoroglaena* are not plentiful enough, all the sequences of the new species are clearly positioned in the *Psoroglaena* group. Whilst a closely related genus *Agonimia* was analyzed with the new species in the molecular phylogeny, lack of the DNA sequences for *Phylloblastia*, another comparable genus, did not allow for the classification in the phylogeny. However, the taxonomy of the new species was diagnosed by comparing with *Phylloblastia* in ecology and morphology. All species in *Phylloblastia* are known as foliicolous, if the corticolous *P. gyeongsangbukensis* is disregarded, and the new corticolous species in *Psoroglaena* can be distinguished from *Phylloblastia*.

A few molecular studies have previously been accomplished on *Psoroglaena* at the family level [9,10]. The genus *Psoroglaena* was classified on its own clade without any closely related genus in both studies. *Psoroglaena* is solely located in a clade without having any other genera. Although *Psoroglaena* is close to *Agonimia* or *Phylloblastia* in morphology, *Agonimia* was located far from *Psoroglaena* in their phylogeny, and *Phylloblastia* was not concerned in the phylogeny due to the lack of DNA data. This study explains that *Psoroglaena* is classified as distinct in Verrucariaceae as well. Although the LSU tree shows *Psoroglaena* is positioned close to *Agonimia* and *Flakea*, the ITS and mtSSU trees show that our genus is located on its own clade and unique in the family. Interestingly, *P. abscondita* in Muggia et al. [9] is classified far from other *Psoroglaena* species as the species is weirdly positioned in the LSU tree in this study. They decided that the distinct position of *P. abscondita* is due to the difference in the ribosomal DNA. We also experienced such a difference and the LSU locus showed high variability from *Pyrenodesmia* [36] and *Arthonia* (a manuscript in preparation). The high variability of the LSU sequences does not allow a clear classification in molecular phylogeny.

Five species of *Psoroglaena* were previously reported in Korea, which includes *P. chirisanensis*, *P. coreana*, *P. gangwondoensis*, *P. japonica*, and *P. sunchonensis* [18,19,20,21,22]. Corticolous species are *P. chirisanensis*, *P. gangwondoensis*, and *P. sunchonensis*, and other two are saxicolous. Here, the corticolous species are further considered to identify the new species. Both *P. chirisanensis* and *P. gangwondoensis* differ from the new species by inhabiting bark of *Robinia psuedoacasia* L. and representing larger ascospores without any filamentous projections on the thallus. *Psoroglaena sunchonensis* inhabits a similar substrate (i.e., *Salix alba* L.) and shows minute projections on thallus as the new species have such filamentous projections locally. Although sharing above similarities in ecology and morphology, the former represents whitish ascomata, different from constantly pale-yellow ones of the latter, and much wider (8–9 μm wide) and muriform ascospores unlike the purely transversely septate spores of the latter. All recorded Korean species of *Psoroglaena* have no DNA sequence data, and were not analyzed in molecular phylogeny.

*Psoroglaena* species can be collected by chance in the field as they are so minute and can be easily confused with something like algae or bryophytes. Particularly, Korean *Psoroglaena* species were found on planted trees just nearby a stream flowing through a local city (e.g., *P. sunchonensis*), or on a common tree species, *Robinia pseudoacacia* (e.g., *P. chirisanensis* and *P. gangwondoensis*), which is one of the representative invasive species to a disturbed area by artificial change. The new species was collected in a small swamp just below a forest road of a local mountain. Further fortuitous collections in an urban or a disturbed area for *Psoroglaena* and following molecular results from them will clarify their taxonomy beyond morphology.

## Figures and Tables

**Figure 1 jof-08-00392-f001:**
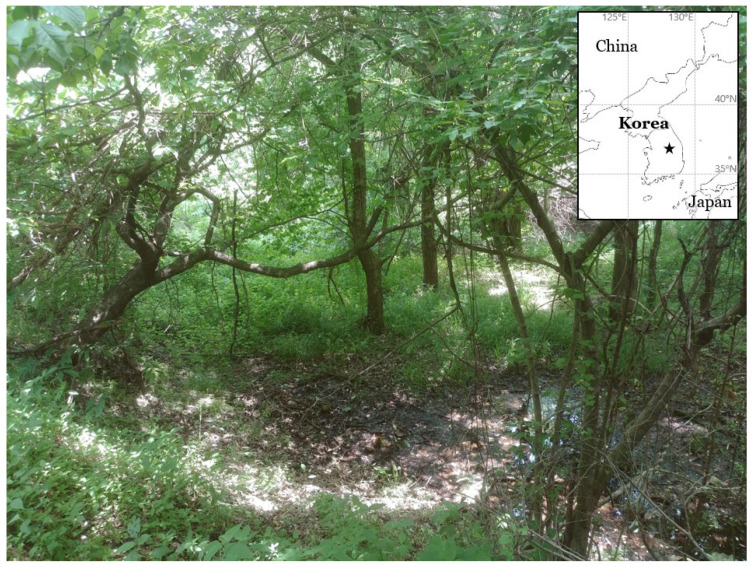
Specific collection site for the new species *Psoroglaena humidosilvae* (black star mark). The small swamp is located below a forest road and comprised mainly of *Salix koreensis* Andersson, *Alnus incana* (L.) Moench subsp. *hirsuta* (Turcz. ex Spach) Á.Löve and D.Löve, *Fraxinus sieboldiana* Blume and *Weigela subsessilis* (Nakai) L.H.Bailey. Lichens were detected on the barks of *Salix koreensis* and *Alnus incana* subsp. *hirsuta*.

**Figure 2 jof-08-00392-f002:**
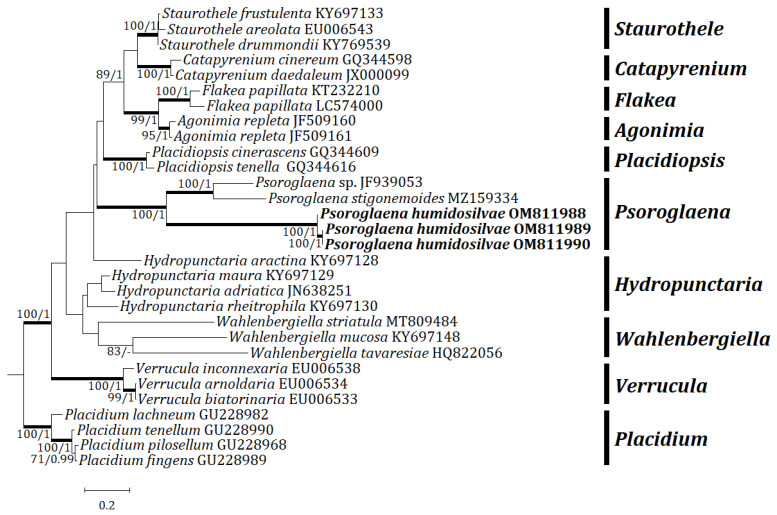
Phylogenetic relationships among available species in the genus *Psoroglaena* based on a maximum likelihood analysis of the dataset of ITS sequences. The tree was rooted with the sequences of the genus *Placidium* based on Muggia et al. [9]. Maximum likelihood bootstrap values ≥70% and posterior probabilities ≥ 95% are shown above internal branches. Branches with bootstrap values ≥90% are shown as fatty lines. The new species, *P. humidosilvae*, is presented in bold as their DNA sequences were produced from this study. All species names are followed by the GenBank accession numbers.

**Figure 3 jof-08-00392-f003:**
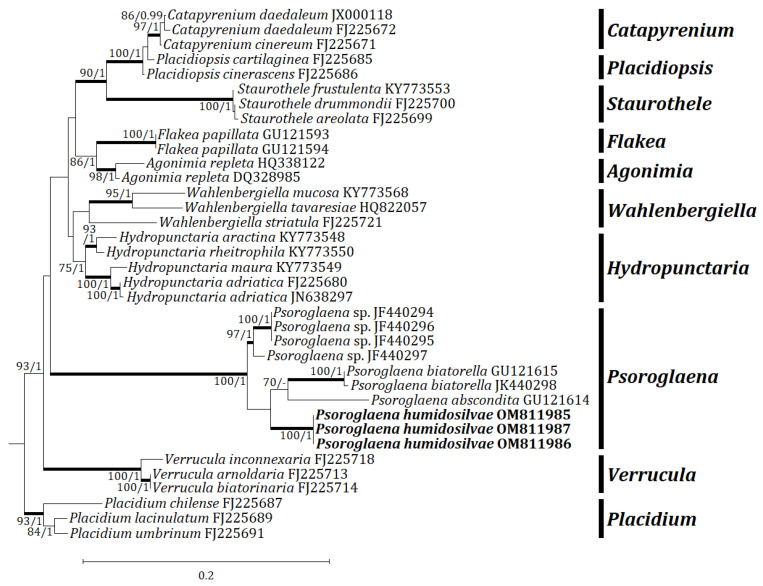
Phylogenetic relationships among available species in the genus *Psoroglaena* based on a maximum likelihood analysis of the dataset of mtSSU sequences. The tree was rooted with the sequences of the genus *Placidium* based on Muggia et al. [9]. Maximum likelihood bootstrap values ≥70% and posterior probabilities ≥95% are shown above internal branches. Branches with bootstrap values ≥90% are shown as fatty lines. The new species, *P. humidosilvae*, is presented in bold as their DNA sequences were produced from this study. All species names are followed by the GenBank accession numbers.

**Figure 4 jof-08-00392-f004:**
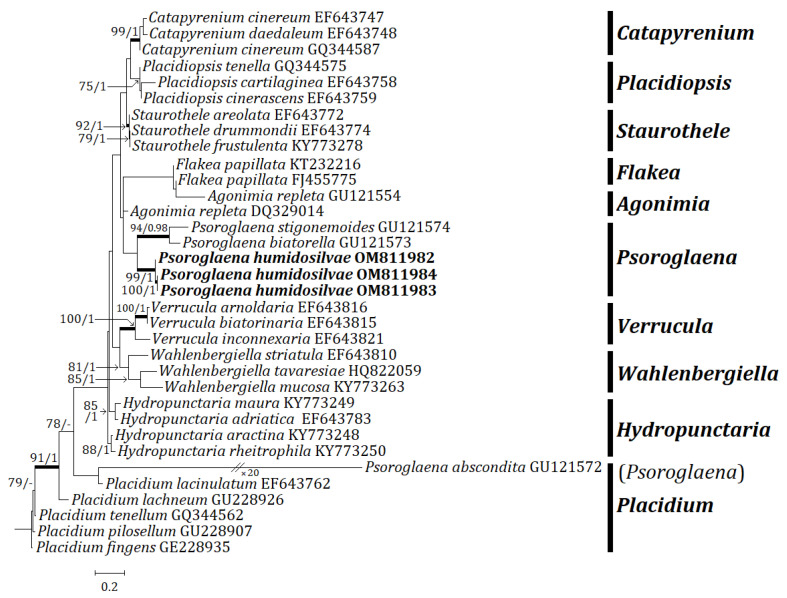
Phylogenetic relationships among available species in the genus *Psoroglaena* based on a maximum likelihood analysis of the dataset of LSU sequences. The tree was rooted with the sequences of the genus *Placidium* based on Muggia et al. [9]. Maximum likelihood bootstrap values ≥70% and posterior probabilities ≥95% are shown above internal branches. Branches with bootstrap values ≥90% are shown as fatty lines. The new species, *P. humidosilvae*, is presented in bold as their DNA sequences were produced from this study. All species names are followed by the GenBank accession numbers.

**Figure 5 jof-08-00392-f005:**
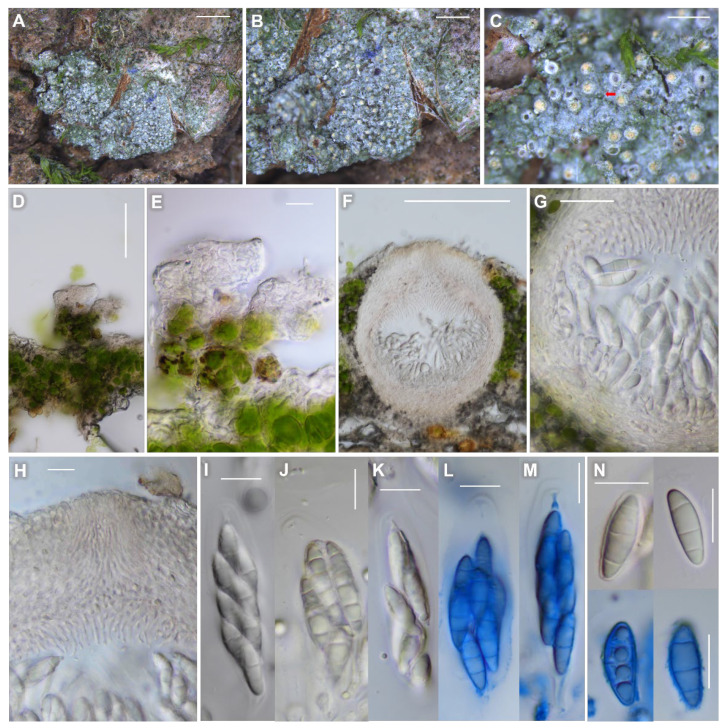
*Psoroglaena humidosilvae* (KBA-L-0002007, holotype for (**A**–**C**), (**F**–**N**); KBA-L-0002005, paratype for (**D**–**E**) in morphology. (**A**–**C**) Habitus and apothecia on bark of *Salix koreensis*. A minute filament (red arrow) shown in (**C**). (**D**–**E**) Appressed filaments (papillae) as minute projections on thallus. (**F**) Globose perithecia without involucrellum. (**G**) The 5- to 6-layered excipulum. (**H**) Ostiolar region with periphyses. (**I**–**M**) Asci clavate to cylindrical. (**N**) Ascospores generally 3-septate. Blue colors stained in Lactophenol cotton blue in (**L**–**N**). Scale bars: (**A**) = 2 mm, (**B**) = 1 mm, (**C**) = 500 μm, (**D**) = 50 μm, (**E**,**H**–**N**) = 10 μm, (**F**) = 100 μm, (**G**) = 20 μm.

**Table 1 jof-08-00392-t001:** Species list and DNA sequence information employed for phylogenetic analysis.

No.	Species	ITS	mtSSU	LSU	Voucher
1	*Agonimia repleta*	-	DQ328985	DQ329014	F 446
2	*Agonimia repleta*	JF509161	-	-	GPN 5935
3	*Agonimia repleta*	JF509160	HQ338122	-	Palice 12970
4	*Agonimia repleta*	-	-	GU121554	Palice 6464
5	*Catapyrenium cinereum*	GQ344598	-	GQ344587	MA16301
6	*Catapyrenium cinereum*	-	FJ225671	EF643747	S. Heidmarsson 2017
7	*Catapyrenium daedaleum*	-	FJ225672	EF643748	C. Gueidan 115
8	*Catapyrenium daedaleum*	JX000099	JX000118	-	Prieto 3051
9	*Flakea papillata*	-	GU121593	-	Perlmutter 1267
10	*Flakea papillata*	KT232210	-	KT232216	T. Tonsberg 34186
11	*Flakea papillata*	LC574000	-	-	TNS:YO12329
12	*Flakea papillata*	-	GU121594	FJ455775	Perlmutter 1263
13	*Hydropunctaria adriatica*	-	FJ225680	EF643783	C. Gueidan 669
14	*Hydropunctaria adriatica*	JN638251	JN638297	-	NMW < GBR_:C.2011.014.61
15	*Hydropunctaria aractina*	KY697128	KY773548	KY773248	DUKE:LA31123
16	*Hydropunctaria maura*	KY697129	KY773549	KY773249	AMNH:LA31903
17	*Hydropunctaria rheitrophila*	KY697130	KY773550	KY773250	AMNH:LA30737
18	*Placidiopsis cartilaginea*	-	FJ225685	EF643758	C. Gueidan 458
19	*Placidiopsis cinerascens*	-	FJ225686	EF643759	C. Gueidan 585
20	*Placidiopsis cinerascens*	GQ344609	-	-	MA161305
21	*Placidiopsis tenella*	GQ344616	-	GQ344575	Ll 281352
22	*Placidium chilense*	-	FJ225687	-	O. Breuss (DUKE, s.n.)
23	*Placidium fingens*	GU228989	-	GU228935	LI 271006
24	*Placidium lachneum*	GU228982	-	GU228926	M. Prieto 322
25	*Placidium lacinulatum*	-	FJ225689	EF643762	K. Knudsen 733
26	*Placidium pilosellum*	GU228968	-	GU228907	M. Prieto 439
27	*Placidium tenellum*	GU228990	-	GQ344562	MA16300
28	*Placidium umbrinum*	-	FJ225691	-	O. Breuss (LI, 14.461)
29	*Psoroglaena abscondita*	-	GU121614	GU121572	JN1742
30	*Psoroglaena biatorella*	-	GU121615	GU121573	Hafellner 64389
31	*Psoroglaena biatorella*	-	JF440298	-	KRAM-51901
**32**	** *Psoroglaena humidosilvae* **	**OM811988**	**OM811985**	**OM811982**	**KBA-L-00002005**
**33**	** *Psoroglaena humidosilvae* **	**OM811989**	**OM811986**	**OM811983**	**KBA-L-00002007**
**34**	** *Psoroglaena humidosilvae* **	**OM811990**	**OM811987**	**OM811984**	**KBA-L-00002008**
35	*Psoroglaena* sp.	JF939053	-	-	KRAM:L-64117
36	*Psoroglaena* sp.	-	JF440296	-	KRAM:L-64117 (KRAP 13/4)
37	*Psoroglaena* sp.	-	JF440295	-	KRAM:L-64117 (KRAP 13/9)
38	*Psoroglaena* sp.	-	JF440294	-	KRAM:L-64117 (KRAP 13/5)
39	*Psoroglaena* sp.	-	JF440297	-	KRAM:L-64117 (KRAP 18/8)
40	*Psoroglaena stigonemoides*	-	-	GU121574	-
41	*Psoroglaena stigonemoides*	MZ159334	-	-	K(M):98005
42	*Staurothele areolata*	EU006543	FJ225699	EF643772	C. Gueidan 378
43	*Staurothele drummondii*	KY769539	FJ225700	EF643774	C. Gueidan 831
44	*Staurothele frustulenta*	KY697133	KY773553	KY773278	DUKE:Heidmarsson 2066
45	*Verrucula arnoldaria*	EU006534	FJ225713	EF643816	C. Gueidan 594
46	*Verrucula biatorinaria*	EU006533	FJ225714	EF643815	C. Roux 24443
47	*Verrucula inconnexaria*	EU006538	FJ225718	EF643821	C. Roux 22721
48	*Wahlenbergiella mucosa*	KY697148	KY773568	KY773263	AMNH:LA31918
49	*Wahlenbergiella striatula*	-	FJ225721	EF643810	C. Gueidan 688
50	*Wahlenbergiella striatula*	MT809484	-	-	FH:BHI-F1163
51	*Wahlenbergiella tavaresiae*	HQ822056	HQ822057	HQ822059	C. Gueidan 1101
	**Overall**	**30**	**36**	**34**	

DNA sequences which were generated for the new species *Psoroglaena humidosilvae* in this study, are presented in bold. All others were obtained from GenBank. The species names are followed by GenBank accession numbers and voucher information. ITS, internal transcribed spacer; mtSSU, mitochondrial small subunit; LSU, nuclear large subunit ribosomal RNA; Voucher, voucher information.

**Table 2 jof-08-00392-t002:** Comparison of the new species with close species in the genus *Psoroglaena*.

Species	*Psoroglaena humidosilvae*	*Psoroglaena abscondita*	*Psoroglaena chirisanensis*	*Psoroglaena stigonemoides*
Projections on thallus	locally a few projections	not observed	not observed	minute projections developed, which form minutely fruticose thallus
Perithecia color	pale yellow	pale brown	pale yellow to pale yellow-brown	pale brown
Perithecia (mm in diam.)	0.1–0.2	c. 0.1	c. 0.2	0.2–0.4
Perithecia shape in section	globose	globose	–	ovoid-obpyriform
Exciple thickness (μm)	c. 20	c. 10	25–30	25–30
Asci (μm)	30–65 × 10–20	45–50 × 10–15	c. 40 × 12	80–100 × 10–12 *
Ascospore septation	(1-)3-septate	1–3-septate	(1-)3-septate	3–4-(5-)septate
Ascospores (μm)	12–18 × 4.5–6.5	12–20 × 3.5–4.5	18–20 × 5.5–6.5	16–21 × 5–6 13–23 × 4–6 *
Ascospore constriction at septa	generally not constricted, or slightly constricted for 1-septate spores	not constricted	distinctly constricted	slightly constricted
Substrate	bark (*Salix koreensis*)	bark (*Sambucus*, *Juniperus*)	bark (*Robinia pseudoacasia*)	bark (*Sambucus*, *Acer*, *Quercus*, *Salix, Ulmus*); moss; leaf
Reference	KBA-L-0002000 (paratype), KBA-L-0002005 (paratype), KBA-L-0002006 (paratype), KBA-L-0002007 (holotype)	[34,35]	[18]	[4,34]

The morphological and ecological characteristics for the closely related species are referenced from the previous literature. The measurements with asterisk marks in the column of *P. stigonemoides* are referenced from Lücking [4], which describes the characteristics of foliicolous *P. stigonemoides* possibly. All information on the new species is produced from selected specimens (KBA-L-0002000, KBA-L-0002005, KBA-L-0002006, KBA-L-0002007) in this study.

**Table 3 jof-08-00392-t003:** Key to the species of *Psoroglaena*.

1.	On rock	**2**
–	On bark, moss or leaf	**4**
2.	Perithecia colorless, 0.1–0.2 mm diam.; goniocyst absent; ascospores 18–22 × 5.5–6.5 μm	** *P. coreana* **
–	Perithecia orangish to dark brown, 0.2–0.4 mm diam.; goniocyst present; ascospores 11–17 × 5–6.5 μm	**3**
3.	Perithecia orange-pink; goniocyst 20–40 μm diam.; ascospores generally 1-septate (rarely 3-septate), 11.5–13.5 × 5.5–6.5 μm	** *P. infossa* **
–	Perithecia pale orange-brown to dark brown; goniocyst 10–20 μm diam.; ascospores 3-septate, 13–17 × 5–6 μm	** *P. japonica* **
4.	On moss or leaf	**5**
–	On bark	**14**
5.	On moss	**6**
–	On leaf	**8**
6.	Thallus inconspicuous, not minutely fruticose; perithecia less than 0.2 mm diam.; particularly growing on *Radula flaccida*	** *P. hepaticicola* **
–	Thallus crustose to minutely fruticose; perithecia over 0.2 mm diam.	**7**
7.	Thallus generally granular-verrucose, with soredia-like granules; perithecia 0.2–0.6 mm diam.; ascospores 3-septate (young) to submuriform, 28–38 × 10–14 μm; sometimes growing on humus-rich ground	** *P. biatorella* **
–	Thallus generally minutely fruticose, locally with soredia-like granules; perithecia 0.2–0.4 mm diam.; ascospores transversely 3–5-septate, 16–21 × 5–6 μm	** *P. stigonemoides* **
8.	Thallus appressed filamentous to microsquamulose; photobiont cells in distinct, uni- or biseriate filaments	**9**
–	Thallus crustose, smooth or minutely arachnoid-tomentose by fungal hyphae (not algal filaments!); photobiont cells in short, irregular threads or irregular plates to almost solitary	**12**
9.	Thallus microsquamulose with erect, irregularly branched, bi- or triseriate filaments; ascospores transversely 3–5-septate	** *P. stigonemoides* **
–	Thallus appressed filamentous with regularly branched to almost unbranched, uni-or rarely biseriate filaments; ascospores (as far known) submuriform to muriform	**10**
10.	Thallus with punctiform soralia	** *P. sorediata* **
–	Thallus lacking soralia	**11**
11.	Thallus rather large (up to 100 mm across), composed of mostly parallel, very long and rarely branched, uniseriate filaments, pale olive-green	** *P. epiphylla* **
–	Thallus smaller (up to 30 mm across), composed of richly branched, uni- or biseriate filaments forming rounded rosettes, vivid green	** *P. ornata* **
12.	Thallus minutely arachnoid-tomentose; perithecia with algal layer between excipulum and (secondary) involucrellum	** *P. arachnoidea* **
–	Thallus smooth, perithecia lacking distinct algal layer	**13**
13.	Thallus continuous, without prothallus, lacking isidia, perithecia pale yellow; ascospores 3-septate, 16–24 × 3–4 μm	** *P. perminuta* **
–	Thallus dispersed into irregular patches on dark prothallus, with scattered, disc-shaped isidia; perithecia dark brownish gray to black; ascospores 3–5-septate, 22–27 × 5–7 μm	** *P. laevigata* **
14.	Thallus filamentous to microsquamulose	**15**
–	Thallus crustose, leprose without elongated or branched projections	**20**
15.	Perithecia not seen; thallus minutely fruticose, with dense cortical papillae (c. 2 × 1 μm)	** *P. spinosa* **
–	Perithecia present; thallus generally minutely fruticose or granular with locally soredia-like granules	**16**
16.	Ascospores submuriform to muriform	**17**
–	Ascospores transversely septate only	**18**
17.	Ascospores 60–70 × 15–20 μm	** *P. costaricensis* **
–	Ascospores 14–18 × 8–9 μm	** *P. sunchonensis* **
18.	Ascospores 7-septate, 18–24 μm long	** *P. cubensis* **
–	Ascospores 3–5-septate, 12–21 μm long	**19**
19.	Thallus minutely fruticose with heavy projections; perithecia pale brown, ovoid-obpyriform, 0.2–0.4 mm diam.; asci 80–100 × 10–12 μm; ascospores 3–5-septate, 16–21 × 5–6 μm	** *P. stigonemoides* **
–	Thallus smooth to granular with locally minute projections; perithecia pale yellow, globose, 0.1–0.2 mm diam.; asci 32–65 × 10–20 μm; ascospores (1-)3-septate, 12–18 × 4.5–6.5 μm	** *P. humidosilvae* **
20.	Ascospores muriform	**21**
–	Ascospores transversely septate only	**22**
21.	Ascospores up to 18 × 6 μm	** *P. dictyospora* **
–	Ascospores 22–29 × 7–9 μm	** *P. gangwondoensis* **
22.	Perithecia dark olive-brown to dull black; ascospores (5-)7-septate	** *P. halmaturina* **
–	Perithecia pale yellow to pale brown; ascospores (1-)3-septate	**23**
23.	Thallus leprose; perithecia 80–120 μm wide; exciple 10–12 μm thick; ascospores 3.5–4.5 μm wide	** *P. abscondita* **
–	Thallus granular to verrucose; perithecia 170–180 μm wide; exciple 25–30 μm thick; ascospores 5.5–6.5 μm wide	** *P. chirisanensis* **

## Data Availability

Publicly available datasets were analyzed in this study. The multilocus data alignment file was deposited in TreeBASE (http://www.treebase.org, accessed on 3 March 2022; accession number 29486). All newly generated sequences were deposited in GenBank (https://www.ncbi.nlm.nih.gov/genbank/, accessed on 25 February 2022; Table 1). All new taxa were deposited in MycoBank (https://www.mycobank.org/, accessed on 11 February 2022).

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
