# Peer review of "A New Lichenized Fungus, Psoroglaena humidosilvae, from a Forested Wetland of Korea, with a Taxonomic Key to the Species of Psoroglaena"

_jof, 2022, doi:10.3390/jof8040392_

Round 1
Reviewer 1 Report
This article describes a new species of Psoroglaena from Korea, based on morphological, ecological and molecular data. It is limited in scope (description of one species) but it presents novel data of interest to lichen taxonomists and contributes to the knowledge on the lichen flora from Korea. It is well written, but would benefit from the editing of a native English speaker. Methods are relatively sound and illustrations, character table, identification key and description are of good quality and will be very helpful to readers. Upon revision of issues highlighted below, I would recommend this article for publication.
- there are few places where the English could be improved (eg, change "The Psoroglaena" by "Psoroglaena" on lines 21, 25, 27). Please have the manuscript reviewed by an English native speaker.
- the introduction failed to mention any previous phylogenetic work on Verrucariaceae and Psoroglaena. Yet, most of the taxa included in the generated trees came from these previous studies (eg, Gueidan et al. 2007, Muggia et al. 2010, etc...). These references need to be mentioned and added to the text.
- 2.3 Phylogenetic analyses section: you need to mention from which publications most of the sequences you are using come from.
- lines 74-76: it sounds like your final alignments, after removing missing and ambiguous regions are 2338 (ITS), 968 (mtSSU), 2404 (LSU) long. That sounds very unlikely. It sounds more like the length of your alignments before you exclude any characters (introns, ambiguous regions, etc...). Please clarify that. Also, you are mentioning removing parsimony un-informative characters, but you are not using parsimony, but maximum likelihood and Bayesian. That doesn't make sense. Please use all unambiguous characters.
- in the legend of figure 2, you are mentioning using Placidium as an outgroup. What is it based on? If it is based on a previous phylogenetic study, please mention it. Same for the outgroup for other genes. Always mention where you got that information from.
- line 126-127: this is unfortunately a contaminating sequence. You can leave it in your tree and then mention that it is likely a contaminant, or remove it from your tree.
- line 166: "... globose, prominent (20-30 µm high)..." What are you giving measurement from? If it is the perithecia, it's probably a mistake (they can't be this size). Or is it the ostiole? Please clarify, thanks!
- line 215: remove extra full stop after "table 3".
- discussion: the discussion could be expended by referring to the literature on the phylogenetic placement of this genus and the relationships among genera of Verrucariaceae.
Reviewer 2 Report
jof-1643489
Dear Editor,
A very interesting study that describe “A new lichenized fungus, Psoroglaena humidosilvae, from a forested wetland of Korea, with a taxonomic key to the species of Psoroglaena” with morphological and molecular data. The study is generally important due to that authors does include a taxonomic key to the species of Psoroglaena.
As stated earlier, the manuscript is well structured and conducted by the authors. The authors demonstrated with consistent morphological and molecular data the description of new specie in the study.
I have minor comentaris
______________________________________________________________________
Line 95: Table 1. Species list and DNA sequence information employed for phylogenetic analysis
The authors should include the author of the scientific name for lichens species of Psoroglaena genus for the first time in the text.
Line 215: Table 3. . Key to the species of Psoroglaena.
Delete duplicate point “.”, in table title
- Perithecia less than 0.2 mm diam.; ascospores 18–22 × 5.5–6.5 μm P. coreana
– Perithecia over 0.2 mm diam.; ascospores 11–17 × 5–6.5 μm
In the key, I think that the perithecia size is not good character because perithecia size is very vaiable. Thus if is applicable I susggets include other characteristics.
Finally, I think that results and discussion can be written as a combined section
Yours sincerely,
